# CLEARER: Multi-Scale Neural Architecture Search for Image Restoration

**Yuanbiao Gou**
College of Computer Science
Sichuan University, China
gouyuanbiao@gmail.com

**Boyun Li**
College of Computer Science
Sichuan University, China
liboyun.gm@gmail.com

**Zitao Liu**
TAL Education Group
Beijing, China
liuzitao@100tal.com

**Songfan Yang**
TAL Education Group
Beijing, China
songfan.yang@qq.com

**Xi Peng**[*]
College of Computer Science
Sichuan University, China
pengx.gm@gmail.com

## Abstract

Multi-scale neural networks have shown effectiveness in image restoration tasks, which are usually designed and integrated in a handcrafted manner. Different from the existing labor-intensive handcrafted architecture design paradigms, we present a novel method, termed as multi-sCaLe nEural ARchitecture sEarch for image Restoration (CLEARER), which is a specifically designed neural architecture search (NAS) for image restoration. Our contributions are twofold. On one hand, we design a multi-scale search space that consists of three task-flexible modules. Namely, 1) Parallel module that connects multi-resolution neural blocks in parallel, while preserving the channels and spatial-resolution in each neural block, 2) Transition module remains the existing multi-resolution features while extending them to a lower resolution, 3) Fusion module integrates multi-resolution features by passing the features of the parallel neural blocks to the current neural blocks. On the other hand, we present novel losses which could 1) balance the tradeoff between the model complexity and performance, which is highly expected to image restoration; and 2) relax the discrete architecture parameters into a continuous distribution which approximates to either 0 or 1. As a result, a differentiable strategy could be employed to search when to fuse or extract multi-resolution features, while the discretization issue faced by the gradient-based NAS could be alleviated. The proposed CLEARER could search a promising architecture in two GPU hours. Extensive experiments show the promising performance of our method comparing with nine image denoising methods and eight image deraining approaches in quantitative and qualitative evaluations. The codes are available at https://github.com/limit-scu.

## 1 Introduction

Due to the complicated environments and the quality of digital image acquisition devices, the captured images are often contaminated by various of signal-dependent or -independent noises. To recover the clean image from the observed noisy image, a huge number of image restoration methods [30, 1, 5, 10] have been proposed based on different handcrafted priors. However, once the prior is inconsistent with the real data distribution, an unpleasant recovery will be achieved.

---

[*]Corresponding author

Different from the prior-based methods, some deep learning based methods adopt a data-driven fashion to remove the possible corruptions by mapping the degraded images to the latent clean versions, which have achieved state-of-the-art performance. In other words, these so-called learning-based methods substitute the explicit and handcrafted image priors with implicit and learning-based priors which are captured by neural architectures [41, 11, 29]. Hence, it is considerably important to seek an effective neural network architecture for facilitating image restoration. As a matter of fact, the advances of image restoration in recent years are benefited from the developments of various handcrafted neural network architectures [27, 14, 15, 19, 24, 40, 31, 34].

In these handcrafted models, multi-scale architectures [16, 37, 33, 35, 39, 42] have played a significant role in improving the performance, of which the basic idea is to fuse features with different resolutions/scales. The success of multi-scale methods are attributed to different roles of low- and high-resolution networks. In brief, the low-resolution networks could capture the global structure of the given image, while losing the perception of details. In contrast, the high-resolution networks could preserve the local details of images, while being with less semantics and robustness to noise. As either using low- or high-scale information alone cannot guarantee encouraging recovery, it is highly expected to use them together so that their merits are taken and the demerits are overcome.

Although it is well-known that multi-resolution fusion is helpful to boosting image recovery performance [7, 12, 44, 23], it is difficult to obtain an effective architecture and almost all existing works rely on human design. Such a handcrafted paradigm has suffered from the following limitations. First, it is labor-intensive to seek an effective architecture, while the image recovery performance is sensitive to neural architecture according to the advances in recent years. Especially, multi-scale networks often consist of multiple subnetworks, which further increases the difficulty of handcrafted design. Second, one more daunting task of multi-scale architecture design is unknown when to fuse multi-scale features. Third, as multi-scale network is more complex than the single-scale one, it is highly expected to find an elegant tradeoff between the model complexity and recovery quality. Clearly, it is difficult to achieve the above goals through human design.

To overcome the aforementioned limitations, we propose a novel method, termed as multi-sCaLe nEural ARchitecture sEarch for image Restoration (CLEARER), which could be one of the first attempts towards automatic integration and design of multi-resolution neural architectures. The contributions and novelty could be summarized as follows:

- We propose a multi-resolution search space consisting of three task-flexible modules, *i.e.*, parallel module, transition module, and fusion module. All these modules are specifically designed to facilitate image restoration. With the increasing depth of our CLEARER, a series of high-to-low resolution subnetworks (cells) are gradually added one by one.

- We propose employing a data-driven strategy to search when to fuse low- and high-resolution features with the help of our novel loss function. The proposed method is time efficient, which only takes two hours to search architectures using a single V100 GPU.

- We show a feasible solution to formulate the model complexity into our loss function in a differentiable manner. In other words, our method could control the balance between performance and model size, which is highly expected to a wide range of applications including resource-constrained scenario. In addition, to avoid the trivial solution caused by the continuous relaxation using softmax, an architecture loss is proposed which could polarize the distribution of architecture parameters, *i.e.*, approach to 0 or 1.

## 2   Related Works

This work is close to image restoration and neural architecture search (NAS) which are briefly introduced in this section.

### 2.1   Image Restoration

To date, a number of image restoration algorithms have been proposed, which achieved remarkable development in numerous practical applications [19, 40, 15, 21]. For instance, [19] proposed incorporating non-local operations into a recurrent neural network for image restoration. [24] proposed using two residual connections to exploit the potential of paired operations for image

restoration. [40] recovered clean image by constructing a memory block using a recursive unit and a gate unit. [33] introduced a multi-scale convolutional neural network for image dehazing and achieved state-of-the-art performance. [23] fused the multi-scale information using a grid-like network and employs attention mechanism to improve the dehazing performance. [47] proposed a multi-stream densely connected network to efficiently leverages features from different scales for image deraining. Besides these explicit multi-scale methods, the models with skip connections [9] could also be regarded as using multi-scale information in an implicit way.

Different from these handcrafted architectures, our method could automatically construct a multi-resolution network, which could remarkably alleviate the difficulty in architecture design while providing a feasible way to balance model complexity and performance.

## 2.2  Neural Architecture Search

NAS aims to automatically discover desirable neural architectures [43, 3, 50, 49, 32] by using one of the following search strategies, namely, evolutionary algorithm (EA), reinforcement learning (RL), gradient-based methods, etc. In the early period, some works [3, 32] adopted EA to search architectures, which obtained the best architecture via the iterative crossovers and mutations of population. Different from EA, RL-based methods, such as Q-learning [49] and policy gradients [50], trained a recurrent neural network which acts as a controller to generate architectures by traversing a predefined search space. Despite the promising performance of these two families of methods, they often face the explosion problem of architecture combination, which is computationally inefficient. To alleviate this issue, recent focus has shifted to gradient-based methods, such as DARTS [20] and BDA [22]. The basic idea of the gradient-based NAS is to relax the discrete and non-differentiable architecture representation to a continuous and differentiable surrogate, thus allowing the efficient search of architectures using gradient descent. Motivated by the competitive performance and search efficiency of the gradient-based NAS, we also employ a differentiable manner to search our multi-scale architecture representation but with a significant difference. In brief, we aim to search for a super-network instead of a cell. Moreover, we alleviate the trivial solution issue caused by using softmax to perform differentiable relaxation [20] via our architecture loss.

To the best of our knowledge, there are only two studies have been conducted towards developing NAS for image restoration, *i.e.*, E-CAE [38] and HiNAS [46]. However, our method is remarkably different from them in the following aspects. First, these pioneers did not utilize specific characteristics of image restoration, whereas our method is a task-specific NAS. More specifically, the proposed CLEARER could automatically utilize and fuse the multi-resolution features that are highly expected to image restoration. Second, the search space is different. In details, both E-CAE and HiNAS design their search space using some basic operators such as conv$3 \times 3$, whereas we design a higher-level search space considering specific characteristics of multi-resolution fusion. Third, these methods did not consider the resource-constrained scenario, whereas our CLEARER could control the trade-off between model complexity and performance. Fourth, both HiNAS and our CLEARER are differentiable. However, HiNAS aim at searching a cell which is further repeatedly stacked to construct the whole network, whereas our CLEARER directly search a super-network consists of different cells while keeping computational efficiency. As pointed out in [2], although cell-based NAS could remarkably reduce the time and space cost, it will achieve inferior performance.

## 3  Multi-scale NAS for Image Restoration

The proposed CLEARER consists of five different components as shown in Figure 1.(c). In brief, parallel module, transition module, fusion module, cell, and convolutional layer. In the super-network, the used two parallel modules keep the original resolution, which are used to receive the input noisy images. The fusion module is used to integrate the multi-resolution features, followed by a convolutional layer. The high-to-low resolution network will be added after each transition module and the nearby cell is with learnable structure. Note that, different from most of existing gradient-based NAS [20, 46], the cells of CLEARER are probably with different rather than the same structure. In other words, we directly search a super-network which consists of $S$ learnable cells. In the following, we will elaborate the proposed multi-resolution search space, the used differentiable search strategy, and the loss functions.

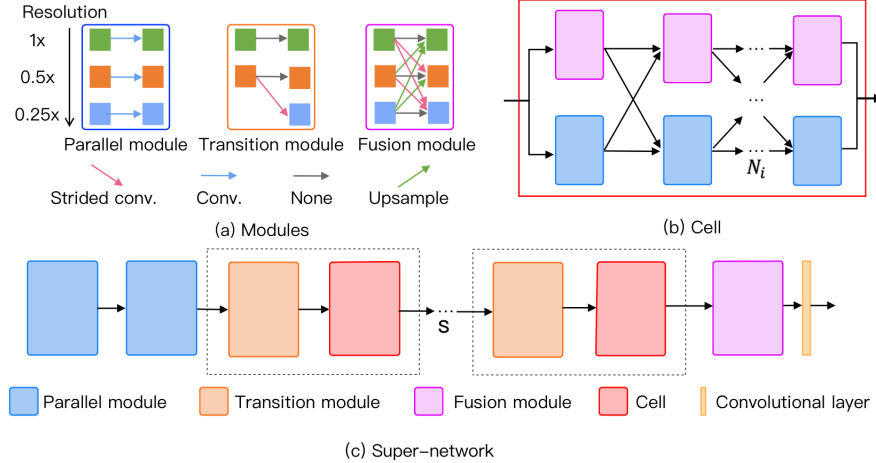

Figure 1: An illustration to our multi-scale search space. In brief, the proposed CLEARER uses modules to build the cell, and then use cells to build the super-network. a) The proposed multi-scale search space which consists of three types of modules, *i.e.*, parallel module, transition module, and fusion module. The colored blocks denote the features with different resolutions and we take three scales of resolution as a showcase in this example. To be specific, the resolution reduced from 1x to 0.5x to 0.25x from top to bottom. b) The structure of cell which contains the Parallel or the Fusion module at each column, and each cell consists $N_i$ columns. At each column, our differentiable search strategy will automatically seek the best module, *i.e.*, our CLEARER will search when to fuse the multi-resolution features through the fusion module. c) The whole structure of our super-network. After each Transition module, the resolution will be reduced by 0.5.

## 3.1 Differentiable Multi-Scale Search Space

As shown in Figure 1.(a), our multi-scale search space mainly composes of three basic modules, *i.e.*, parallel module, transition module, and fusion module. The parallel module connects multiple neural blocks in parallel, and the resolution of features in each parallel line remains unchanged via the convolution operation. The transition module will add a lower-scale resolution network via the strided convolution and simultaneously keep the resolution in the horizontal direction via none operation. The fusion module fuses multi-resolution features through the strided convolution, upsampling, and none operator.

With the parallel module and the fusion module, we design a cell as illustrated in Figure 1.(b). In details, each cell consists of $N_i$ columns, and each column is a parallel module or fusion module, where $i = \{1, 2, \cdots, S\}$ is the index of cell. By taking one cell as a showcase, we illustrate the optimization process in Figure 2.

For a noisy input, the proposed CLEARER will directly pass it through two parallel modules, $S$ pairs of the transition module and cell, and a fusion module followed by a $1 \times 1$ convolutional layer. Such a progressive architecture will gradually add the high-to-low resolution networks into the super-net so that the resolution will decrease and the final output could encapsulate multi-resolution information. For each search, we need to determine the architectures of $S$ cells and $N_i$ columns, thus the size of our search space is $2^{N_1+N_2+\cdots+N_S}$.

Let $y_j$ be the input to the $j$-th column of a cell and $\{f_j^p(\cdot), f_j^f(\cdot)\}$ be the parallel module and the fusion module at the $j$-th column. Hence, we have

$$y_{j+1} = \alpha_j^p f_j^p(y_j) + \alpha_j^f f_j^f(y_j), \tag{1}$$

where $\alpha_j^p = 0, \alpha_j^f = 1$ or $\alpha_j^p = 1, \alpha_j^f = 0$, *i.e.*, the search space is discrete. To efficiently search a desirable architecture, we employ a continuous relaxation of $\{\alpha_j^p, \alpha_j^f\}$, *i.e.*, the above binary constraint is replaced by $\{\alpha_j^p, \alpha_j^f\} \in (0, 1)$ and $\alpha_j^p + \alpha_j^f = 1$. Although our CLEARER adopts the softmax to achieve the above relaxation like [20], we further constrain the distribution of the

architecture parameter $\{\alpha_j^p, \alpha_j^f\}$ to approximate the discrete distribution as elaborated in the next section.

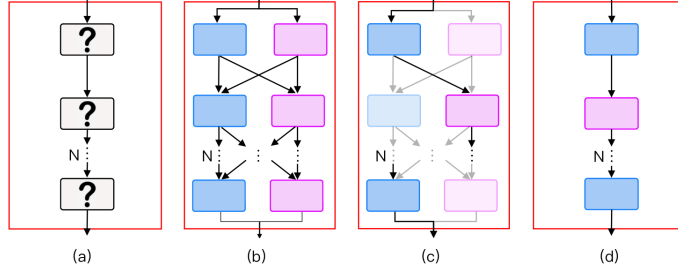

(a)      (b)      (c)      (d)

Figure 2: An illustration to the optimization of one cell: (a) The modules in our cell are initially unknown. (b) Continuous relaxation of the search space by placing a mixture of candidate modules. (c) Iterative optimization of the mixing architecture parameters and the network weights. (d) Obtaining the final architecture from the learned mixing probabilities. Note that, we directly search a super-network which consists of $S$ learnable cells. Here, we only take one cell as an example.

## 3.2 Loss Function

To search for a desirable architecture for image restoration, we propose the following loss function:

$$\mathcal{L} = \mathcal{L}_{Res} + \lambda_1 \mathcal{L}_{Arch} + \lambda_2 \mathcal{L}_{Comp}, \tag{2}$$

where $\mathcal{L}_{Res}$ is the restoration loss between the ground-truth clean image and the recovered clean image of CLEARER. $\mathcal{L}_{Arch}$ is the architecture regularization loss which is used to constrain the distributions of architecture parameters to close 0 or 1. $\mathcal{L}_{Comp}$ is the differentiable loss for measuring the model complexity. The nonnegative parameters $\lambda_1$ and $\lambda_2$ are used to balance the corresponding items.

For a given noisy input $x$, $\mathcal{L}_{Res}$ aims to minimize the mean squared error (MSE) between the corresponding ground-truth $\hat{x}$ and the recovered clean image $f(x)$. Mathematically,

$$\mathcal{L}_{Res} = \frac{1}{T} \sum_{i=1}^{T} (f(x_i) - \hat{x}_i)^2 \tag{3}$$

where $T$ is the number of $x$'s pixels.

Although the above continuous relaxation makes the search space differentiable, there exists a problem caused by the relaxation using the softmax. To be specific, it will lead to that both $\alpha^p$ and $\alpha^f$ are around 0.5, i.e., the different candidate modules are indistinguishable (see our supplementary material), which is inconsistent with our original goal and formulation. To solve this problem, we propose $\mathcal{L}_{Arch}$ which is enforced onto all architecture parameters with the following formulation,

$$\mathcal{L}_{Arch} = -\frac{1}{N} \sum_{a \in \{\alpha^p, \alpha^f\}} (\alpha \log \alpha + (1 - \alpha) \log(1 - \alpha)) \tag{4}$$

where $N = 2 \sum_{i=1}^{S} N_i$ is the number of architecture parameters. This regularization will enforce $\alpha$ to distribute to approach either 0 or 1.

One major favorite of NAS to image restoration is controllable model complexity which is important to a variety of resource-constrained scenarios such as mobile phone. However, this property has not been touch in existing works [46, 38] due to the formulation in the model complexity, as well as the non-differentiability. In this work, we formulate the model complexity of the cells into $\mathcal{L}_{Comp}$ as below:

$$\mathcal{L}_{Comp} = \frac{1}{N} \sum (\alpha^p \mathcal{C}^p + \alpha^f \mathcal{C}^f) \tag{5}$$

where $\mathcal{C}^p$ and $\mathcal{C}^f$ are the complexity of the parallel module and fusion module, respectively. The complexity could be predefined using module size, time cost, FLOPs, and so on. In the following experiments, we simply use the module size to measure the complexity.

Like [20], we adopt a bilevel optimization paradigm to iteratively update architecture parameters via $\mathcal{L}$ and network weights via $\mathcal{L}_{Res}$ using two non-overlapping data partitions. Namely, $\mathcal{L}_{Res}$ is used both in the optimization of architecture parameters and network weights. $\mathcal{L}_{Arch}$ and $\mathcal{L}_{Comp}$ are the architecture regularizations, which are only used to optimize the architecture parameters.

## 4 Experiments

To demonstrate the effectiveness of our method, we carry out experiments on two image restoration tasks, *i.e.*, image denoising and deraining[2]. In the following, we first introduce the experimental settings and then show the qualitative and quantitative results on some public datasets. Finally, we perform ablation study and parameter analysis to our model. Due to space limitation, we present more experimental results and experimental details such as network architectures and parameter settings in the supplementary material.

### 4.1 Experimental settings

In this section, we elaborate the experimental configurations for architecture search and model training. For the evaluations, two popular metrics are used in quantitative comparisons, i.e., PSNR and SSIM.

**Architecture Search Settings:** The super-network that we build for denoising contains three cell and each cell consists of four cascade modules, *i.e.*, $S = 3$ and $N_i = 4(i = 1, ..., S)$ (see Fig. 1). Namely, the size of our search space is $2^{12}$. Following [19, 24, 27, 40, 46], we utilize the training set and validation set from BSD500 to train and find the best neural architecture with the highest performance. After that, we use the well-trained network to process the testing images and report the corresponding results.

We adopt the standard SGD optimizer with the momentum of 0.9 and the weight decay of 0.0003 to optimize the parametric model. The learning rate automatically decays from 0.025 to 0.001 via the cosine annealing strategy [25]. To optimize the architecture parameters, we adopt Adam [13] optimizer with the learning rate of 0.0003, and the weight decay of 0.001. In the search process, we build a data batch with the size of 32 by randomly cropping patches of $64 \times 64$ from training images, and feed the batch to the network with the maximal iteration of 10,000. To avoid trivial solution, only the weight parameters are updated in the first 1,000 iterations. Namely, the weight parameters and architecture parameters will be alternatively updated after the 1,000-th iteration. For fair comparisons, we simply set $\lambda_1 = 0.01$ and $\lambda_2 = 0$ by ignoring the model complexity like the compared approaches. In the ablation study, we will show the effectiveness of this parameter in balancing the model complexity and recovery performance.

All two image recovery tasks adopt the same configurations and their only one difference lies on the data distribution. Like the compared methods such as [19], we employ data augmentation techniques including rotation and flip to augment the data during search, training and testing.

**Model Training Settings:** After finishing architecture search, we train the obtained network for 100,000 iterations with the batch size of 32 and patch size of 64. To optimize the network parameters, we employ the Adam optimizer with the learning rate decays from 0.01 to 0 via the cosine annealing strategy.

### 4.2 Comparisons on Image Denoising

**Data Sets:** We carry out denoising experiments on three datasets, *i.e.*, BSD500 [28], BSD68 [48], and Set12. In details, BSD500 consists of 500 natural images of which 200, 100, and 200 images are used for training, validation, and testing, respectively. BSD68 includes 68 different natural images and Set12 contains 12 different scenes, which are also used to test the model trained on the BSD500 training set. By following [46, 4, 31, 24, 40], the noisy images are generated by adding white Gaussian noises to the clean images with three corruption levels, *i.e.*, $\sigma = 30, 50, 70$.

**Compared Methods:** For comprehensive comparisons, we compare the proposed CLEARER with nine representative denoising methods. In details, BM3D [4], WNNM [8], RED [27], MemNet [40],

NLRN [19], DuRN-P [24], N3Net [31], HiNAS [46], and E-CAE [38]. Note that, the first seven approaches are traditional handcrafted architectures and the last two are NAS-based models. For a comprehensive study, we evaluate CLEARER on both the image patches and the whole images, denoted by CLEARER-P and CLEARER.

**Results:** Tables 1–2 report the results on BSD500 comparing with seven handcrafted denoising methods and two NAS based methods, respectively. In addition, Table 3 demonstrates the results on BSD68 and Set12 with the noise level $\sigma = 50$ (See Table 3). Note that, we do not report the performance of E-CAE and HiNAS on BSD68 and Set12 because the corresponding results are unavailable. Figure 3 shows some denoising examples and more visual comparisons have been presented in the supplementary material.

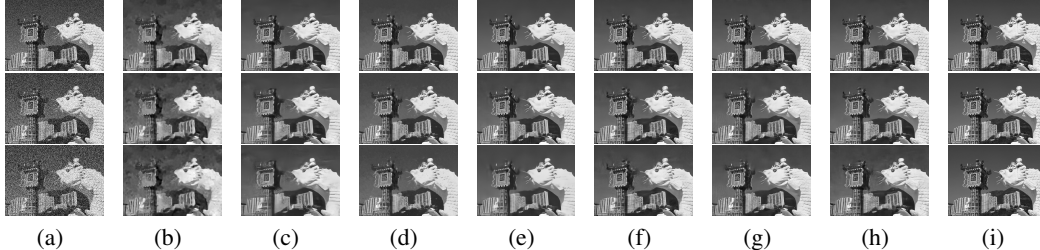

| (a) | (b) | (c) | (d) | (e) | (f) | (g) | (h) | (i) |

Figure 3: A qualitative comparison on BSD500. The noise levels from the top to the bottom are $\sigma = 30, 50, 70$, respectively. From the left to the right are Input, BM3D, RED, WNNM, NLRN, DuRN-P, N3Net, CLEARER, and Ground truth. Note that, we did not obtain the encouraged visual result of E-CAE in our experiments.

Table 1: Denoising comparisons with handcrafted methods on the BSD500.

| Methods | $\sigma = 30$ | | $\sigma = 50$ | | $\sigma = 70$ | |
|---|---|---|---|---|---|---|
| | PSNR | SSIM | PSNR | SSIM | PSNR | SSIM |
| BM3D | 27.31 | 0.7755 | 25.06 | 0.6831 | 23.82 | 0.6240 |
| RED | 27.95 | 0.8056 | 25.75 | 0.7167 | 24.37 | 0.6551 |
| WNNM | 27.48 | 0.7807 | 25.26 | 0.6928 | 23.95 | 0.3460 |
| MemNet | 28.04 | 0.8053 | 25.86 | 0.7202 | 24.53 | 0.6608 |
| N3Net | 28.66 | 0.8220 | 26.50 | 0.7490 | 25.18 | 0.6960 |
| NLRN | 28.15 | 0.8423 | 25.93 | 0.7214 | 24.58 | 0.6614 |
| DuRN-P | 28.50 | 0.8156 | 26.36 | 0.7350 | 25.05 | 0.6755 |
| CLEARER | 28.54 | 0.8203 | 26.40 | 0.7465 | 25.06 | 0.6894 |
| CLEARER-P | 29.68 | 0.8439 | 27.49 | 0.7768 | 26.09 | 0.7267 |

Table 2: Denoising performance of three NAS-based methods on the BSD500. The time includes the cost for architecture searching and model training.

| Methods | $\sigma = 30$ | | $\sigma = 50$ | | $\sigma = 70$ | | GPU | Time cost (hours) | Search |
|---|---|---|---|---|---|---|---|---|---|
| | PSNR | SSIM | PSNR | SSIM | PSNR | SSIM | | | |
| HiNAS | 29.14 | 0.8403 | 26.77 | 0.7635 | 25.48 | 0.7129 | 1 Tesla V100 | 16.50 (2.54x) | gradient |
| E-CAE | 28.23 | 0.8047 | 26.17 | 0.7255 | 24.83 | 0.6636 | 4 Tesla P100 | 96.00 (14.77x) | EA |
| CLEARER | 28.54 | 0.8203 | 26.40 | 0.7465 | 25.06 | 0.6894 | 1 Tesla V100 | 6.50 | gradient |
| CLEARER-P | 29.68 | 0.8439 | 27.49 | 0.7768 | 26.09 | 0.7267 | 1 Tesla V100 | 6.50 | gradient |

Table 3: Denoising performance (PSNR/SSIM) on Set12 and BSD68 with the noise level of $\sigma = 50$.

| Datasets | BM3D | RED | WNNM | MemNet | NLRN | N3Net | DuRN-P | CLEARER | CLEARER-P |
|---|---|---|---|---|---|---|---|---|---|
| Set12 | 26.55/0.7423 | 27.32/0.7748 | 27.05/0.7775 | 27.36/0.7791 | 27.64/0.7980 | 27.43/- | 27.14/0.7829 | 27.43/0.8021 | 28.08/0.8129 |
| BSD68 | 25.60/0.6866 | 26.29/0.7124 | 25.87/0.6982 | 26.34/0.7190 | 26.47/0.7298 | 26.39/- | 26.31/0.7276 | 26.31/0.7352 | 27.25/0.7681 |

As shown in the tables, our method outperforms most manually-designed and NAS-based methods. Specifically, our method is 0.31 and 0.0156 higher than E-CAE, 0.04 and 0.0047 higher than DuRN-P

in terms of PSNR and SSIM when noise level $\sigma = 30$. Comparing with HiNAS, although relatively inferior results are obtained, our method is remarkably efficient than E-CAE and HiNAS for searching and training thanks to our multi-scale search space and gradient-based search strategy. To be specific, our multi-scale search space consists of some modularized operators rather than basic operators, which is specifically designed for image restoration. Therefore, although our CLEARER searches a super-network instead of motifs, it is more efficient than E-CAE and HiNAS. Similar results could also be observed when $\sigma = 50$ and $\sigma = 70$.

## 4.3 Comparisons on Image Deraining

**Data Sets:** To show the effectiveness of our method in image deraining, we carry out evaluations on Rain800 [47] dataset which contains 700 synthesized training images and 100 synthesized test images. For a fair comparison, we randomly sample 100 images from the training images for validation and use the remaining 600 images for training. All the test images are used for testing.

**Compared Methods:** In experiments, we compare our CLEARER with eight state-of-the-art deraining models, including DSC [26], LP [18], DetailsNet [6], JORDER [45], JORDER-R [45], SCAN [17], RESCAN [17], and HiNAS [46].

**Results:** From Table 4, one could observe that our method and HiNAS remarkably outperforms the other tested methods. For example, CLEARER is 3.12 and 0.0275 higher than the best handcrafted method in PSNR and SSIM, respectively. Although equally good result is obtained in SSIM, CLEARER is 0.9 higher than HiNAS in PSNR. The results again show the effectiveness of our method.

Table 4: Deraining results on Rain800.

| Methods | DSC | LP | DetailsNet | JORDER | JORDER-R | SCAN | RESCAN | HiNAS | CLEARER |
|---------|-----|-----|-----------|--------|----------|------|--------|-------|---------|
| PSNR | 18.56 | 20.46 | 21.16 | 22.24 | 22.29 | 23.45 | 24.09 | 26.31 | 27.21 |
| SSIM | 0.5996 | 0.7297 | 0.7320 | 0.7763 | 0.7922 | 0.8112 | 0.8410 | 0.8685 | 0.8685 |

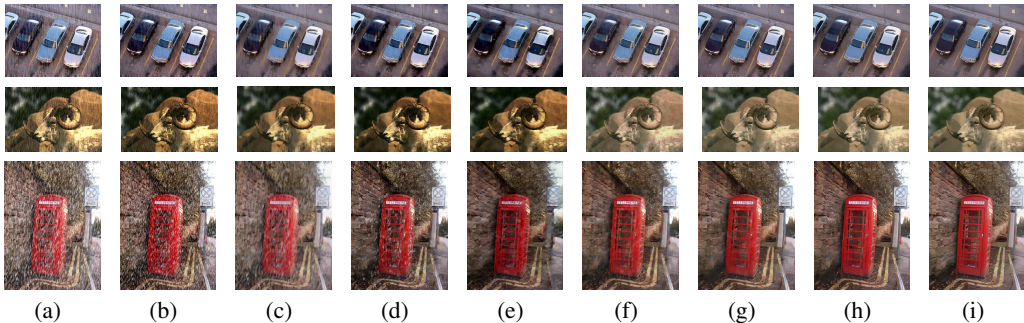

|     |     |     |     |     |     |     |     |     |
|-----|-----|-----|-----|-----|-----|-----|-----|-----|
| (a) | (b) | (c) | (d) | (e) | (f) | (g) | (h) | (i) |

Figure 4: The qualitative comparisons on Rain800. From the left to the right are Input, DSC [26], LP [18], DetailsNet [6], JORDER [45], SCAN [17], RESCAN [17], CLEARER, and Ground truth.

## 4.4 Ablation Study and Model Analysis

In this section, we carry out ablation study and model analysis to CLEARER in image denoising task. Our method contains two parameters, *i.e.*, $\lambda_1$ and $\lambda_2$. In brief, $\lambda_1$ enforces the architecture parameters $\alpha = \{\alpha^f, \alpha^p\}$ close to either 0 or 1. $\lambda_2$ balances the model complexity and performance. From Table 5, one could observe that a larger $\lambda_1$ will lead to a smaller model, better performance, and longer testing time. The possible reason for the smaller model and longer testing time is that a larger $\lambda_1$ will tend to choose the Fusion module which is with fewer parameters and more nonparametric operations. The reason for the better performance is that the architecture regularization $\mathcal{L}_{Arch}$ could widen the gap between the architecture parameters, *i.e.*, $|\alpha^f - \alpha^p|$. As a result, it will increase the distinction between different candidate modules. Comparing with $\lambda_1$, $\lambda_2$ is more significant in control the model size. With the increasing $\lambda_2$, the model size is becoming smaller and the worse results are achieving.

Besides the investigation on $\{\lambda_i\}_{i=1}^2$, we also evaluate the effectiveness of our differentiable search strategy and multi-scale search space by randomly generating a network based on our search space. The experimental configurations are the same as our CLEARER excepted the used search strategy. The new baseline is termed as "CLEARER+Random" as shown in Table 5. From the result, one could see that the PSNR and SSIM reduce to 28.29 and 0.8091, while the model is quite large. This result again shows the effectiveness of our search space and search strategy.

Table 5: Ablation study and parameter analysis on BSD500.

| Configurations | PSNR | SSIM | Testing Time (s) | Parameters (M) |
|---|---|---|---|---|
| $\lambda_1 = 0.00, \lambda_2 = 0.00$ | 28.51 | 0.8187 | 4.28 | 7.25 |
| $\lambda_1 = 0.01, \lambda_2 = 0.00$ | 28.54 | 0.8203 | 5.28 | 6.31 |
| $\lambda_1 = 0.10, \lambda_2 = 0.00$ | 28.53 | 0.8200 | 5.98 | 6.07 |
| $\lambda_1 = 0.01, \lambda_2 = 0.01$ | 28.33 | 0.8100 | 6.70 | 4.59 |
| $\lambda_1 = 0.10, \lambda_2 = 0.10$ | 28.16 | 0.8041 | 5.62 | 4.36 |
| CLEARER + Random | 28.29 | 0.8091 | 4.44 | 7.63 |

## 5   Conclusion

In this paper, we propose a novel NAS method which is specifically designed for image restoration in a differentiable manner. In brief, we design a multi-resolution search space with three task-flexible and interpretable modules that are favorite to the task. In addition, we propose a novel loss function which shows a feasible solution to control the model complexity and performance that is highly expected to the resource-constrained scenarios. The proposed CLEARER adopts a differentiable way to directly search a super-network and only two Tesla V100 GPU hours are taken. Extensive experiments on two image restoration tasks show the promising performance of our CLEARER comparing with 17 state-of-the-art approaches. In the future, we will further improve our method and explore its potential in more low-/high-level vision tasks.

## Broader Impact

The proposed method is a specifically designed neural architecture search (NAS) method for image restoration. Namely, two areas, NAS and image restoration, are involved. Image restoration is a common topic in low-level vision tasks which aims to restore the clean image from the degraded one and mainly used to improve the quality of digital images. NAS aims to automatically design the high-performance neural architectures and has been applied to many vision tasks. Therefore, we will discuss the impacts of our method from both aspects.

First, there is a risk of removing some necessary degradations during recoveries, such as watermark, subtitle, and mosaic. Namely, such a technology has the potential of prejudicing the rights of others with improper use. Second, NAS could help people to search for an effective neural architecture for some specific tasks. This technology saves a lot of labors and greatly reduces the domain expertise required in the manual design process. However, NAS would further intensify the black-box nature of deep neural networks which brings tremendous security risks when applied to some critical fields such as autopilot and medical. In addition, to search for a desirable architecture, a lot of energy will be consumed, while causing massive $CO_2$ emissions. For example, using NAS to find a specialized neural network and train it from scratch for each case, which causes $CO_2$ emission as much as five cars' lifetime [36].

## Acknowledgements

This work was supported in part by NFSC under Grant U19A2081, 61625204, and 61836006; in part by the Fundamental Research Funds for the Central Universities under Grant YJ201949; in part by the Fund of Sichuan University Tomorrow Advancing Life; in part by the Spark Project of Sichuan University under Grant 2018SCUH0070; and in part by Beijing Nova Program (Z201100006820068) from Beijing Municipal Science & Technology Commission.

## Footnotes

[2]Note that: rain streaks could be regarded as one signal-dependent noise.

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
