[Supplementary Material]

# Supplementary Material for "CLEARER: Multi-Scale Neural Architecture Search for Image Restoration"

**Yuanbiao Gou**
College of Computer Science
Sichuan University, China
gouyuanbiao@gmail.com

**Boyun Li**
College of Computer Science
Sichuan University, China
liboyun.gm@gmail.com

**Zitao Liu**
TAL Education Group
Beijing, China
liuzitao@100tal.com

**Songfan Yang**
TAL Education Group
Beijing, China
songfan.yang@qq.com

**Xi Peng**[*]
College of Computer Science
Sichuan University, China
pengx.gm@gmail.com

## 1 Experiments

In this material, we present more details of our CLEARER, including module configurations, searching architectures, the distribution of architecture parameters. In addition, we also report additional experimental comparisons in terms of the denoising and the deraining tasks.

### 1.1 Module Configurations

In the paper, we present a multi-scale search space which is casted into a differentiable supernet consisting of three modules, *i.e.*, parallel module, transition module, and fusion module. In this subsection, we elaborate the architectures of these three modules. As shown in Figure 1.(a), there are four operation sequences in the modules, namely, *strided convolution* (Strided conv.), *Convolution* (Conv.), *Upsample*, and *None*. Except for the *None* operation, the others are the combination of some basic operations as shown in Figure 1.(b)-(d).

### 1.2 Architecture Parameters

As mentioned in the main body of the paper, the super-network we build for restoration contains three cells and each cell consists of four cascade modules. Namely, there are 12 cascade modules in total. Each module could be either parallel module or fusion module, which is determined by optimizing the architecture parameters $\alpha^p$ and $\alpha^f$. To polarize the distribution of $\alpha^p$ and $\alpha^f$, the architecture loss $\mathcal{L}_{Arch}$ is proposed with the parameter $\lambda_1$. Here, we show the influence of different $\lambda_1$ w.r.t. the distribution of architecture parameters. The numerical values of architecture parameters with different $\lambda_1$ are shown in Table 1 and the corresponding scatter plot is shown in Figure 2.

From the table and the figure, one could observe that both $\alpha^p$ and $\alpha^f$ are around 0.5 for all cascade modules when $\lambda_1 = 0$. Namely, there is no obvious gap between the best module and the other module. When $\lambda_1 = 0.01$, this gap is widened and a larger gap is achieved by $\lambda_1 = 0.1$. Therefore, with the increase of $\lambda_1$, the distribution of architecture parameters tend to the polarization distribution, *i.e.*, $\alpha^p$ approaches to 0 when $\alpha^f$ approaches 1, and vice versa. Such a polarization distribution is more consistent with the original discrete search space and in favor of distinguishing the best module,

---

[*]Corresponding author

Figure 1: An illustration to the intra architecture of our search space. (a) The three types of modules which form our multi-scale search space. Here, we take a three-scale resolutions with the downsampling factor of 0.5 as a showcase. Note that the colored square in the modules denotes the feature with different resolutions and the colored edge represents the sequence of operations. (b) The strided convolution is used to down sample features. It is composed of $R_1$ pairs of *Convolution* and *BatchNorm* operations, followed by a *ReLU* operation. Here, $R_1$ is automatically determined by the ratio between the target resolution $T_{res}$ and the source resolution $S_{res}$ via $R_1 = \log_{0.5} \frac{T_{res}}{S_{res}}$. (c) The convolutional sequence is arranged in a residual manner for each parallel direction. In brief, it is in order of *Convolution*, *BatchNorm*, *ReLU*, *Convolution*, *BatchNorm*, and *ReLU*. (d) Upsample sequence transits the resolution of features from low to high, which consists of $R_2$ sequences of *Convolution*, *BatchNorm*, and *Upsample*. Similarly, $R_2$ is also automatically determined by the ratio between the target resolution $T_{res}$ and the source resolution $S_{res}$ via $R_2 = \log_2 \frac{T_{res}}{S_{res}}$. Note that, we will change the number of channel with the resolution. Specifically, the number of channel will be doubled if the resolution is halved, In our evaluations, we empirically set the number of channel to 32 in the highest resolution direction.

Table 1: Distribution of architecture parameters with different $\lambda_1$. $\alpha^p$ and $\alpha^f$ are the architecture parameters of parallel module and fusion module, respectively. $\{1, \cdots, 12\}$ is the index of modules.

| $\lambda_1$ | Arch | The index of module. | | | | | | | | | | | |
|---|---|---|---|---|---|---|---|---|---|---|---|---|---|
| | | 1 | 2 | 3 | 4 | 5 | 6 | 7 | 8 | 9 | 10 | 11 | 12 |
| 0.00 | $\alpha^p$ | 0.4603 | 0.4474 | 0.4958 | 0.4873 | 0.5079 | 0.5020 | 0.5084 | 0.4934 | 0.5086 | 0.4969 | 0.5098 | 0.5113 |
| | $\alpha^f$ | 0.5397 | 0.5526 | 0.5042 | 0.5127 | 0.4921 | 0.4980 | 0.4916 | 0.5066 | 0.4914 | 0.5031 | 0.4902 | 0.4887 |
| 0.01 | $\alpha^p$ | 0.3663 | 0.5411 | 0.4771 | 0.5144 | 0.4975 | 0.5123 | 0.4909 | 0.4090 | 0.5015 | 0.4971 | 0.4892 | 0.5299 |
| | $\alpha^f$ | 0.6337 | 0.4589 | 0.5229 | 0.4856 | 0.5025 | 0.4877 | 0.5091 | 0.5910 | 0.4985 | 0.5029 | 0.5108 | 0.4701 |
| 0.10 | $\alpha^p$ | 0.1601 | 0.8286 | 0.8343 | 0.1650 | 0.1635 | 0.1638 | 0.1658 | 0.1658 | 0.8393 | 0.1665 | 0.1643 | 0.8378 |
| | $\alpha^f$ | 0.8399 | 0.1714 | 0.1657 | 0.8350 | 0.8365 | 0.8362 | 0.8342 | 0.8342 | 0.1607 | 0.8335 | 0.8357 | 0.1622 |

Figure 2: An illustration to the distribution of the architecture parameters with different $\lambda_1$. As shown in the figure, the distribution of each $\{\alpha^p, \alpha^f\}$ pair satisfies $\alpha^p + \alpha^f = 1$. When $\lambda_1 = 0$, the average gap between $\{\alpha^p, \alpha^f\}$ pairs is about 0.0278, *i.e.*, $\frac{1}{12}\sum_{j=1}^{12}|\alpha_j^p - \alpha_j^f| = 0.0278$, where $j$ is the index of cascade modules. Such a gap is too small to well distinguish the best module from the other candidates. However, this gap will be widened to 0.0620 when $\lambda_1 = 0.01$ and 0.6709 when $\lambda_1 = 0.1$ which make the candidate modules more distinguishable.

thus alleviates the latent issues caused by the continuous relaxation and improves the performance of learned architectures.

## 1.3 Searching Architectures

In this section, we present the detailed architectures learned for denoising and deraining. As shown in Figure 3 and Figure 4, the obtained architectures achieve multi-resolution in a progressive manner. In the vertical direction, there are at most four resolutions and each is denoted as a kind of colored square. In the horizontal direction, after each transition module, there are four cascade modules which need to be learned.

From the figures, one could find that there are more fusion modules than parallel modules in both the denoising and the deraining architectures. Specifically, the learned two architectures both contain eight fusion modules and four parallel modules, and the only one difference between them is the position of the fusion and the parallel modules. From the observations, we could conclude that: 1) the multi-scale information is remarkably important to image restoration. 2) different tasks require fuse the multi-resolution at different positions, which indicates the importance of NAS.

Figure 3: The architecture learned for image denoising. The colored squares denote the features with different resolutions and the colored edges represent the operation sequences which are illustrated in the Figure 1.

## 1.4 Qualitative Comparisons

Besides the result presented in the main body of our submission, we show more qualitative comparisons on denoising and deraining here. Figure 5–6 demonstrate the denoising results on the BSD500 dataset, and Figure 7 illustrates the deraining results on the Rain800 dataset. According to the results,

Figure 4: The architecture learned for image deraining.

one could find that our proposed CLEARER achieves pleasant visualization results both in denoising and deraining.

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

Figure 5: The qualitative comparisons on the denoising task using BSD500. From the top to the bottom for each image, the noise levels are $\sigma = 30, 50, 70$. From the left to the right are Input, BM3D [1], RED [9], WNNM [3], NLRN [6], DuRN-P [7], N3Net [10], CLEARER, and Ground truth.

|  (a) | (b) | (c) | (d) | (e) | (f) | (g) | (h) | (i) |

Figure 6: The qualitative comparisons on denoising task using BSD500. The noise levels for each image from the top to the bottom are $\sigma = 30, 50, 70$. From the left to the right are Input, BM3D [1], RED [9], WNNM [3], NLRN [6], DuRN-P [7], N3Net [10], CLEARER, and Ground truth.

Figure 7: The qualitative comparisons on deraining task using Rain800. From the left to the right are Input, DSC [8], LP [5], DetailsNet [2], JORDER [11], SCAN [4], RESCAN [4], CLEARER, and Ground truth.

(a)    (b)    (c)    (d)    (e)    (f)    (g)    (h)    (i)