[Reviews · NeurIPS 2020]

Review 1

Summary and Contributions: This paper presents a method to search a multi-scale network for image restoration using the Neural Architecture Search (NAS). Moreover, the authors propose a loss function to relax the discrete architecture parameters into a continuous distribution. Experiments are performed for image denoising and image deraining.

Strengths: The paper is easy to follow. The proposed method is faster than competing NAS based image restoration approaches.

Weaknesses: 1: Limited novelty: CLEARER uses multi-scale search space that consists of three types of modules: parallel module, transition module, and fusion module. All of these modules were originally proposed in [2, 1].The authors did not cite these works when mentioning the said modules throughout the paper. 2: The proposed method generates two different architectures: one for image denoising and the other for image deraining. It seems inconvenient, as for every new task we would have a different architecture. it would have been more desirable to aim for a universal network that can be trained for different image restoration tasks. 3: The authors claim several times that CLEARER is particularly designed for image restoration. However, they did not provide any analysis/insights of what makes it specific for image restoration. For instance, what makes it suitable for image denoising and image deraining, OR why it would not work for any other applications such as semantic segmentation? 4: Experiments aren’t comprehensive. Only one dataset is used for image deraining evaluation. The reviewer would recommend testing on other commonly used image deraining datasets, such as Rain100L, Rain100H, Rain1200,etc. And for image denoising, the reviewer would recommend testing on real image datasets, such as SIDD and DND. 5: It seems that the architecture generated by CLEARER is driven by the dataset on which it is trained on, rather than the particular image restoration task. The reviewer would suggest to search for an architecture for image deraining using CLEARER, but on a new dataset (not Rain800), and see if it yields the same architecture as in Fig. 4 of the supplementary material. [1] Ke Sun, Bin Xiao, Dong Liu, and Jingdong Wang. Deep high-resolution representation learning for human pose estimation. In Proceedings of the IEEE conference on computer vision and pattern recognition, pages 5693–5703, 2019. [2] Jingdong Wang, Ke Sun, Tianheng Cheng, Borui Jiang, Chaorui Deng, YangZhao, Dong Liu, Yadong Mu, Mingkui Tan, Xinggang Wang, et al. Deep high-resolution representation learning for visual recognition. IEEE transactions on pattern analysis and machine intelligence, 2020.

Correctness: Probably correct.

Clarity: The paper is easy to follow.

Relation to Prior Work: Several references are missing. Please see "Weaknesses".

Reproducibility: Yes

Additional Feedback: Please see "Weaknesses". The issues rasied in "Weaknesses" are desired to be addressed in the rebuttal. Post Rebuttal Comments: The reviewer would first like to thank the authors for addressing the concerns raised in the initial review. The rebuttal has partially addressed the concerns raised in the initial review. However, the proposed method still has some limitations. The PSNR (30.42) and SSIM (14 0.9056) of CLEARER reported on RAIN1400 is inferior to SOTA. The same is also true for the deraining results reported for Rain800 dataset (line 4-5 in the feedback document). Further, even when changing from one deraining dataset to another, the architecture is no longer universaral and changes (line 4-5 and line 12-14). Despite the aforementioned limitations, considering NAS for image restoration tasks has received little attention. This paper is likely to encourage further work in this direction. Therefore, the reviewer would like to raise the final score from 5 to 6.


Review 2

Summary and Contributions: This paper proposes a multi-resolution neural architecture search space with three task-flexible modules for image restoration. Besides, a new loss function is used to balance the model complexity and performance, which is highly expected to a wide range of applications.

Strengths: 1. This paper employs a novel loss function to balance the model complexity and performance that is highly expected to the resource-constrained scenarios. 2. This paper employs a differentiable strategy to search when to fuse multi-resolution features. 3. Comparisons on image denoising and deraining with state-of-the-art approaches show the effectiveness of the proposed CLEARER.

Weaknesses: 1. I am curious that if the proposed method can deal with other ill-posed image restoration tasks, like image deblurring. The authors may want to discuss some of this. 2. How about exchange the block sequence of the parallel module and the transition module? 3. Iterative optimization of the mixing architecture parameters and the network weights seems to be a crucial step. How it influences the final results? 4. The recent work show that some interactions between scales promote the performance of image restoration. Therefore, can the proposed method involves some interactions between scales?

Correctness: Overall, the claims and method are easy to follow, the authors have clearly explained the proposed method.

Clarity: yes, this paper is well written and organized.

Relation to Prior Work: Yes, the authors have clearly discussed the difference, improvements between the proposed method and previous contributions.

Reproducibility: Yes

Additional Feedback: see the weakness == After rebuttal == I am satisfied with the rebuttal, and have no objection for accepting this paper.


Review 3

Summary and Contributions: This paper proposed to solve the image restoration problem in a multi-scale manner, while the model is obtained via network architecture search (NAS), with a structure space constraint loss and a computation amount loss. Extra (lower) scales are incorporated one by one in the pipeline via a transition module. Both quantitative and qualitative results are impressive.

Strengths: + The paper introduces a structure space constraint loss and a computation amount loss, and reports impressive results

Weaknesses: - The contribution in network architecture search is insufficient. - After the first “Fusion module” in the “Cell module”, the features from different scales have been mixed up. Do the authors refer to “multi-scale” as the Parallel module? Or please make it clear if there are any insights. - The loss $\scriptL_{Arch}$ in Eqn. (4) is contrary to the description in the paper, see “Correctness” for more details. - Besides, there are some minor problems. a) Is the structure of the “Transition module” fixed or searched? b) The English should be further polished, and there are some typos. E.g., overcame -> overcome in line 44, consists -> consist in line 141, constraint->constrain in line 160, etc.

Correctness: Although the authors report impressive results, there are technical errors in this paper. As stated in lines 171 and 172, the authors say that “This regularization will enforce $\alpha$ to distribute to approach either 0 or 1”. However, the loss proposed in Eqn. (4) will constrain the value of $\alpha$ to be 0.5, which is contrary to the statement.

Clarity: The English should be further polished. See the weakness.

Relation to Prior Work: Yes

Reproducibility: No

Additional Feedback: The clarification of the loss term and multi-scale have partially addressed my concerns. Considering that NAS for image restoration remains less investigated, I slightly raised my score to encourage this line of work.


Review 4

Summary and Contributions: This paper proposed a multi-scale NAS method for image denoising and deraining. This paper designed three modules and investigated the relationships of these modules. From my viewpoint, this paper substituted for the cells in general NAS methods by the modules for image restorations. In brief, this paper investigated the combination of different designed modules rather than detailed construction of inner cells. From the aspect of design philosophy, this paper introduced three basic modules, i.e., Parallel module, Transition module and Fusion module, which utilized the multi-scale feature representation and multi-resolution features integration and fusion. From the aspect of search strategies, this paper depicted the principles of loss design to reduce the model complexity.

Strengths: The proposed method is relatively similar to DARTS, which is a differential NAS method for high-level vision. Essentially speaking, it is also a cell-searching-based and gradient-based NAS method. In my opinion, it is not a very significant and insightful idea for image restoration. The main contributions are the trade-off training strategy and the loss design strategy.

Weaknesses: 1. For the principles for designed modules, this paper proposed three basic modules for interior image restoration, however, the interior structure of these modules is fixed, which is not so convinced to build these modules. In my opinion, an inner NAS strategy is necessary to search for a considerable structure for image restoration. Furthermore, as shown in Fig.1(c), the global flow is fixed, which consists of two parallel modules and the multiply combination of transition modules and cells. I wonder that the reason that not searching the global connections and demonstration of reasonability. Indeed, theses principled modules are common principles for other vision problems, such as image segmentation and optical flow estimation. Maybe other principled modules, such as attention modules and tasks-specific modules are vital to introduce. 2. From the experiment of this paper, some experiment settings are not very rational. As a significant contribution, the trade-off between model complexity and inference accuracy is not obviously shown when comparing with other methods for image denoising and deraining. In other words, I cannot find the final schemes for these tasks under the consideration of the trade-off between model complexity and inference accuracy. From the supplemented materials, the designed network is very deep. This paper should indicate the parameters of this paper and other networks. From the reproduction aspect, I cannot find how to search for a good structure for deraining. The various rain scenarios are also necessary to demonstrate your performance. Moreover, this paper aims to remove signal-dependent or -independent noises. Some experiments are missing, such as the performance on the real scenarios and RGB colorful images rather than deraining scenarios. 3. As a bilevel optimization method, it is not very clear the relationship between three losses and objective functions (upper and lower formulations). L_{arch} and L_{Comp} should be optimized in the upper functions. 4. Some subjective statements are inappropriate to introduce this paper. Some proofs and references are needed to demonstrate your statement. it is labor-intensive to seek an effective architecture, while the image recovery performance is sensitive to the choice of neural architecture. One more daunting task of multi-scale architecture design is unknown is that when to fuse the multi-scale feature. Besides these explicit multi-scale methods, the models with skip connections [10] could also be regarded as using multi-scale information in an implicit way. (The author should provide a detailed explanation to verify these statements.)

Correctness: As a bilevel optimization method, it is not very clear the relationship between three losses and objective functions(upper and lower formulations).

Clarity: The related works should use the simple past tense.

Relation to Prior Work: Yes

Reproducibility: Yes

Additional Feedback: ---------------------------------------------------------------- After seeing the rebuttal and comments from other reviewers, I agree that the main contribution of this work should not be the bilevel optimization methodology, but the new search space and training strategies for image restoration tasks. From this point of view, I think this paper should have protentional influence for NAS methods in low-level vision areas. Thus, I decide to provide a positive final recommendation for this paper.

[Author Response · NeurIPS 2020]

**R1Q1: Relationship with HRNET:** Our work is inspired by the first work which has been cited in the paper, but we missed the second work which will be reviewed and credited in the revision. Noticed that, HRNET is a handcrafted network, which does not abstract the three modules (i.e., building a search space) for NAS like this paper does.

**R1Q2: A universal architecture for multiple tasks:** As suggested, we conduct the experiments of using the architecture searched for denoising to train for deraining, the PSNR is 26.51 and SSIM is 0.8519 on Rain800.

**R1Q3, R4Q1, R2Q1: Specific characteristics for restoration:** The multi-resolution (MR) has shown effectiveness in image restoration, and all of them are based on the handcrafted design. As it is unclear (also crucial) to know when/where fuse or parallelly extract the resolutions, CLEARER is proposed. It has a MR searching space and MR feature fusion that benefits to image restoration. See L39–44 for more details. Considering the effectiveness of multi-resolution in high-level tasks, we have extended this idea and achieved some encouraging results. In fact, it is exciting to handle various tasks using a unified framework like Wang's work with the development of deep learning.

**R1Q4, R1Q5, R4Q2: Additional experiments on new data:** Due to time limitation, we only obtained the result on RAIN1400 (12600 training images + 1400 testing images). The PSNR and SSIM of our CLEARER are 30.42 and 0.9056. The architecture searched on it has 4 of 12 modules different from the architecture searched on RAIN800.

**R2Q2: Exchange the parallel module and the transition module:** The sequence of the parallel module and the transition module could be exchanged, but correspondingly, the number of parallel subnetworks will automatically change to deal with the input.

**R2Q3: Iteration optimization vs. performance:** Ideally, the architecture parameters are optimized for the converged network weights. Due to the expensive inner optimization, an iterative optimization was proposed as an approximation whose performance and efficiency has been proved by a series of works like DARTS. Noticed that, the major contribution of this work is designing a specific search space (multi-resolution) for image restoration first time.

**R2Q4: Scales interactions:** Yes. On one hand, the proposed fusion module could process cross-scale/-resolution fusion. On the other hand, many residual connections are existing in the parallel modules. Both the two folds provide the interactions between scales.

**R3Q1, R4: Insufficient contributions:** Most of the existing NAS are designed for the classification task, to the best of our knowledge, there is NO specific NAS for image restoration so far. Here, the specific, we refer to multi-resolution interaction which has shown effectiveness in a variety of handcrafted models, e.g., image denoising [29], deraining [37], dehazing [17]. Noticed that, although HiNAS (CVPR'20) could be one of the first NAS for denoising, it does not consider the specific characters of image denoising like our idea. Experiments show the image restoration tasks could remarkably benefit from such our specific design. Is it not valuable and novel to specifically design a NAS for the tasks besides the classification task?

**R3Q2: Do the authors refer to "multi-scale" as the Parallel module?** No, it refers to the multi-resolution features flowing through the network which consists of the parallel, fusion, and transition module. Only all of these together could refer to multi-scale.

**R3Q3+Correctness: The loss $\mathcal{L}_{Arch}$ in Eqn.(4) is contrary/error to the description.** Thanks! We missed a minus in the right-hand side. It should be $\mathcal{L}_{Arch} = -\frac{1}{N}(\sum_{a \in \{\alpha^p, \alpha^f\}} \alpha \log \alpha + (1-\alpha) \log(1-\alpha))$.

**R4Q1: The reason for not searching the inner structure and the global connections:** The reason are two folds. First, prohibitively temporal and spatial costs for searching the whole architecture. In fact, we have attempted to simultaneously search the interior structure, global connection, and modules, but the model is too large even for the 32GB v100 GPU. This problem cannot be solved by adding more GPUs since the model cannot be loaded into different GPUs. Second, this work is not devoted to search for a good interior structure or the global connections, but automatize the integration and design of multi-resolution neural architectures, especially when to fuse or parallelly extract the multi-resolution features. The reason is that multi-resolution has shown effectiveness in a lot of image restoration tasks, but it is crucial and unknown when and where the multi-resolution are fused and higher-resolution is expected.

**R4Q2: Model complexity vs. inference accuracy of other methods:** On one hand, our networks are more complex than other methods due to the multi-resolution architecture which consists of multiple subnetworks. On the other hand, other methods do not explicitly consider this character. In fact, we believe that NAS is promising in real applications of image restoration because the trade-off between model complexity and performance could be precisely controlled.

**R4Q3+Correctness: As a bilevel optimization method, the relationship between three losses and objective functions is unclear.** We would clarify that our contribution is NOT developing a bilevel optimizer, which instead designing a specific searching space for image restoration. In our loss, $\mathcal{L}_{Res}$ is the restoration loss which is used both in the optimization of architecture parameters and network weights. $\mathcal{L}_{Arch}$ and $\mathcal{L}_{Comp}$ are the architecture regularizations, which only used to optimize the architecture parameters.

[Meta-Review · NeurIPS 2020]

The paper initially received mixed ratings, but the reviewers were convinced by the rebuttal. Overall, their assessment is positive; the area chair agrees with their assessment and recommends an accept.